# Validity of Maternal Recall to Assess Vaccination Coverage: Evidence from Six Districts in Zhejiang Province, China

**DOI:** 10.3390/ijerph16060957

**Published:** 2019-03-18

**Authors:** Yu Hu, Yaping Chen, Ying Wang, Hui Liang

**Affiliations:** Institute of Immunization and Prevention, Zhejiang Provincial Center for Disease Control and Prevention, Hangzhou 310000, China; zjmyscyp@163.com (Y.C.); ywang@cdc.zj.cn (Y.W.); hliang@cdc.zj.cn (H.L.)

**Keywords:** vaccination, validity, maternal recall, recall bias, vaccination record

## Abstract

*Background*: Although recall-based data are collected by survey when the vaccination records are not available, the preferred estimates remain the record-based ones due to the limited validity of recall-based data. However, the evidence on validity of maternal recalls is limited and varied across vaccine types. To close the gaps, we validated the maternal recall on vaccination against record-based data in six districts in Zhejiang Province, China. *Methods*: We used a cross-sectional survey of about 648 households with mothers who delivered in the last 12 months prior to the survey in October 2017, from six districts in Zhejiang Province. Vaccination status on five vaccine types scheduled before 12 months of age were collected through maternal recall and vaccination records. The level of agreement and recall bias between the two resources, the sensitivity and specificity of maternal recall were evaluated. Risk factors for maternal recall bias were also identified through logistic regression model for each type of vaccine. *Results*: The level of agreement between recall and record was above 90% across vaccine types, with the recall bias ranged from 2.2% to 9.7%. Recall bias due to over-reporting was slightly higher than that due to under-reporting. Recall bias was positively associated with high parity, home delivery, younger mothers, mothers with low education, and migrant mothers. *Conclusions*: This study indicated most of the vaccination status across vaccine types was accurately identified through maternal recall and supported the use of maternal recall to estimate the vaccination coverage as an alternative in the absence of record-based data.

## 1. Introduction

Vaccination is considered as one of the key public health interventions, in terms of potential health impact and cost-effectiveness, and has been recommended universally since 1970s. In the effort to further reduce mortality and morbidity among children under five, effective and efficient and sustainable provision of childhood vaccination is urgently necessary [1,2,3]. Therefore, it is also important to monitor the progress and evaluate the performance of the vaccination programs such as expanded program on immunization (EPI) through assessing the vaccination coverage by routine or other reliable measurements. Vaccination coverage is a proxy measure of population immunity with relevance to disease control. Since the coverage is closely related to disease incidence, monitoring coverage can identify likely gaps in immunity among target population and room for improvement in EPI [4]. As such, the estimates of vaccination coverage are important for planning, priority setting, and implementing interventions to improve childhood health [5,6,7]. 

There are three main measurements of assessing the vaccination coverage [5]: electronic registry of immunization such as the Zhejiang provincial immunization information system [8], which covers all vaccination clinics in Zhejiang Province and records the demographic and vaccination information of all children under 7 years of age, administrative records like information routinely collected during service delivery, and population-based household surveys such as demographic and health surveys administered in many countries [9,10]. However, each method has advantages and disadvantages. Electronic immunization registries can provide continuous data for coverage monitoring and for management activities such as monitoring vaccine supply and sending vaccination reminders, but their implementation has many challenges and only registered children can be tracked, which will overestimate the vaccination coverage since these children are at greater likelihood of being vaccinated [11]. Vaccination coverage derived from administrative data has the advantage of being routinely available with few funding and human resources, but its reliability has been questioned due to incomplete or inaccurate primary recording of vaccinations, inaccuracy in population denominators, delayed or duplicate reporting, and missing data on national vaccination day campaigns [12]. Household surveys, which need rigorous survey design, sampling, good quality control, and appropriately weighted analyses, are often used as an independent check on estimates from the routine administrative data since EPI managers or global partners do not trust administrative reports. However, household surveys need more technical and financial resource and only cover a sampled population, which should therefore be used judiciously [13].

Vaccination coverage evaluated through household surveys is commonly referred to vaccination records and/or parental recall, especially from mothers. Although data from vaccination records are more reliable than recall, only including written records would be prone to the selection bias by using an inaccurate sampling frame and including recall data that would obviously have information bias. The absence of immunization records is a frequent issue in household survey settings. In 2003, an assessment of the quality of vaccination coverage surveys across 45 countries found that almost one-third of households on average did not have vaccination records [12]. Reports from India [14] and Sudan [15] indicated that there were more than half of the surveyed mothers could not provide vaccination records. A similar result was also found in USA with one-third of parents having no written records on vaccination [16]. Given that, the vaccination coverage estimated through household surveys may be incomplete where vaccination records are missing in practice, and maternal recall data provides an alternative in such cases.

Evidence on the validity of maternal recall for evaluating vaccination coverage in relation to vaccination records is limited, especially in China. Previous studies had demonstrated that parental recall underestimated the vaccination coverage across vaccine types when compared to the administrative data [15,17]. On the other hand, parental recall was found to be relatively consistent in general with slight variation across vaccine types when compared to data from vaccination records [18,19]. However, these studies are dated with no studies being conducted in recent years. To date, there has been no study from east China, where missing vaccination records is a concern in migrant children. Furthermore, previous studies focused on measles containing vaccine (MCV) and diphtheria-tetanus-pertussis combined vaccine (DTP), with limited attention to other vaccines. Since assessing vaccination coverage through household survey is common, it is necessary to evaluate whether maternal recall is valid and can be a supplement measurement for the record-based estimates if vaccination records are not available. Moreover, there has been limited evidence on the risk factors associated with the recall bias for vaccination coverage.

In 2017, a cross-sectional household survey among children aged 12–23 months in six districts of Zhejiang Province was conducted, capturing data from maternal recall and validating it through the vaccination records based on five vaccines that were scheduled before 12 months of age [20]: Bacillus Calmette–Guérin vaccine (BCG), hepatitis B vaccine (HBV), polio vaccine (PV), DTP and MCV. The BCG vaccine and the first doses of HBV are scheduled at birth, and the first dose of OPV is scheduled at two months old, and the first dose of DTP is scheduled at three months old, and the first dose of MCV is administered after eight months. In this study, the validity of vaccination data from parental recall was evaluated and compared to the data from vaccination records in Zhejiang Province, in East China. In addition, the risk factors of recall bias at both the individual and household-level were explored.

## 2. Methods

### 2.1. Study Settings

Zhejiang Province is a developed province located in eastern China, with a total area of 104141 km^2^ and a population of 72 million. Zhejiang’ rapid development had attracted a migrant population of more than 20 million, particularly from the western rural areas of China. Our study was implemented in six districts out of 90 districts in Zhejiang Province. According to the statistics of Zhejiang province in 2016, the total population of Yinzhou, Dinghai, Dongyang, Changxing, Liandu and Kecheng was 884056, 389872, 839515, 632382, 405580 and 438060, respectively.

### 2.2. Subjects and Sampling Procedure

Children aged 12–23 months (born from 31 October 2015 to 30 September 2016) and their mothers were surveyed in October 2017. All target children had the chance to receive the five vaccines when we implemented the field investigation. 

The household-based cluster survey method recommended by WHO was applied in this study [21]. The sample size was calculated based on the following formula: $N=deff\times \frac{z_{\left(1-\frac{\alpha }{2}\right)}^{2}\times p\times \left(1-p\right)}{d^{2}}$. The parameters were set as a two-tailed *α* error of 5%, a permissible error (*d*) of 0.08, a design effect (*deff*) of 2, and the expected coverage of the lowest coverage among the five vaccines mentioned above of 0.9. The final sample size that was required for each district was 108 eligible children, which was divided into 18 children in each of the six cluster (town). The sample size for the entire study was 648. 

The procedure of survey included four steps: first, we applied a probability proportional to population size method to select six towns for every district, with the population size of each town from the census data 2016. Second, we drew lots to choose one community from the selected town. Third, we selected the index household from the selected community using the random number table. Fourth, we visited the adjacent household on the right of the index household when we finished the previous investigation. The remaining 17 households were investigated in the same way. If there were more than one eligible child in household, we drew lots to investigate only one. Households were re-visited if there was somebody living but without any response. If we did not find enough eligible children in the selected community, the closest community was chosen to investigate the remaining children.

### 2.3. Data Collection

Field investigations were conducted at a district level by 30 vaccination staff from the center for disease control and prevention (CDC). However, these vaccination staff were from other ten districts rather than the six districts selected in this study to avoid the selection bias. The investigators needed to attend a half-day training before the investigation. The training focused on the objectives of the survey, the meaning of every item of the questionnaire, and the skill of investigating the sensitive items. 

Characteristics on the socioeconomic status of the household, background characteristics on children and mothers and data on the utilization for maternal and child health care were gathered by a pre-tested questionnaire developed by Zhejiang CDC. Furthermore, all surveyed mothers were required to recall the vaccination status of their child for any dose across the five vaccine types, not through reviewing the vaccination records.

After the questionnaire, investigators should check the vaccination status again through reviewing the vaccination records held by mothers where available or checking that on Zhejiang provincial immunization information system (ZJIIS). Written information on the appropriate vaccine type and number of doses received were recorded. Maternal recall was assessed only in terms of the coverage of vaccine type rather than the number of doses.

### 2.4. Data Analyses

All analyses were performed using STATA 11 (Stata Statistical Software Corp., College Station, TX, USA). All of the analyses were two-tailed, and *p*-values of 0.05 or less were considered to be statistically significant.

The characteristics on the demographic and socio-economic were described, using the proportion for categorical variables. The vaccination coverage of record-based and recall-based was calculated separately across five vaccine types considered. The recall-based and record-based data were compared, assuming the latter to be accurate. In this study, recall bias was defined as the difference in vaccination status between the two resources (i.e., false positives and false negatives). Recall bias was disaggregated into two categories as one was over-reported (recalled as vaccinated while not) or under-reported (recalled as not vaccinated while vaccinated). 

The level of agreement between the estimates from the two resources as well as the sensitivity and specificity of maternal recall were calculated by the two-by-two table (Table 1). The level of agreement was defined as the percentage of mothers accurately recalled the vaccination status of their children [(a + d)/Total]. Sensitivity was defined as the percentage of children whose mothers recalled they were vaccinated and they were vaccinated according to their vaccination records [a/(a + c)]. Specificity was defined as the percentage of children whose mothers recalled they were unvaccinated and they were not vaccinated according to their vaccination records [d/(b + d)]. We used the Kappa statistics (Kappa_-unadjusted_), which was a measure of reliability taking into account the basis of chance, for each type of vaccine as an alternative of the agreement assessment between two resources. A Kappa statistic of ≤0.20 showed poor agreement, 0.21–0.40 fair agreement, 0.41–0.60 moderate agreement, 0.61–0.80 substantial agreement, and 0.81–1.00 almost perfect agreement. The Kappa statistic is commonly affected by the prevalence of an indicator and level of disagreement which leads to a trade-off paradox (high agreement but low Kappa statistic). As such, we used the prevalence and bias adjusted Kappa (Kappa_-adjusted_) to avoid the paradox in this study meanwhile [22,23].

To identify the determinants of maternal recall bias, a series of multivariate logistic regression models by vaccine types were adopted and odds ratio with 95% confidence interval (95% CI). The dependent variable for recall bias took the value of 1 if the two resources disagreed and 0 otherwise for each vaccine. Individual and household characteristics like child age (in months), maternal age (in years), maternal education, occupation, parity, residence (rural/urban), and household income, were included in the model as potential risk factors of recall bias. 

### 2.5. Ethical Considerations

This study was approved by the ethical review board of Zhejiang Provincial Center for Disease Control and Prevention (T-115-D). All methods were carried out in accordance with relevant guidelines and regulations. Written informed consent was obtained from a parent or a legal caregiver of each eligible child enrolled in this study.

## 3. Results

Of the surveyed 648 mothers with eligible children, 68.4% were under 30 years of age, and 81.5% had an education background of senior middle school or above, and 78.2% had fixed jobs, and 50.8% lived in rural areas, and 38.7% were migrant. Besides, 50.2% of the surveyed children were male and 63.7% were the first born. The majority of them (92.1%) were delivered at hospital and 28.2% were from household with a monthly income per capita of >6000 CNY (Table 2).

The coverage based on record and recall was quite similar for each vaccine type. However, the recall data resulted in slightly higher coverage estimates relative to record data. The record-based coverage of BCG was 95.4 while the recall based coverage was 97.0%. The record-based coverage of HBV was 95.8 while the recall based coverage was 96.8%. The record-based coverage of PV was 95.6 while the recall based coverage was 97.0%. The record-based coverage of DTP was 93.8% while the recall based coverage was 97.3%. The record-based coverage of MCV was 94.6% while the recall based coverage was 96.6%. 

The level of agreement between recall and record was above 90% across vaccine types, with the minimal recall bias of 2.2% for BCG and the maximum recall bias of 9.7% for DTP, respectively. Recall bias due to over-reporting was slightly higher than that due to under-reporting. The unadjusted Kappa and the adjusted Kappa both showed perfect agreement between the two data resources with values above 0.7 for all vaccine types (Table 3).

Maternal recall bias for BCG and PV was invariant across socio-demographic characteristics of mothers and children. In contrast, recall bias for HBV was higher for mothers who gave birth at home and for younger mothers. Recall bias for DTP was associated with mothers with high parity, younger mothers, mothers with low education, and migrant mothers. Recall bias for MCV was associated with mothers with high parity, younger mothers, mothers with low education, and migrant mothers (Table 4).

## 4. Discussion 

In this study, we evaluated the validity of maternal recall against vaccination record for assessing the vaccination coverage in six districts, east China. The record-based data was used as reference since they were commonly preferred to recall data in coverage survey worldwide [24,25]. The two data resources were compared and the level of agreement and recall bias were quantified separately. The high coverage rates for each vaccine type were found in both two data resources. The difference in vaccination coverage between the two data resources was minimal with limited recall bias of < 10%. The sensitivity of maternal recall was above 90% in general, while the specificity was around 80%. Risk factors such as parity, place of delivery, maternal age, education background, and immigration status were associated with maternal recall bias of the relevant vaccine types. 

Our results of the slightly overestimated coverage rates of all vaccine types from the recall-based data was similar to findings from previous studies [15,17,25]. It was reported that the overestimation through recall might be associated with the social desirability bias. It meant the maternal recall was sometimes influenced by the concept that childhood vaccination practice was considered socially desirable [26], which came from the vaccination policies such as the school-entry vaccination requirements. The high level of agreement between record- and recall based data found in this study was much similar to the previous reports from other developing countries, where the accuracy of recall was over 80% for MCV, and ranged from 60% to 98% for BCG and DTP [15,18,27]. Another interesting finding was that the level of agreement for BCG and PV was higher than the other vaccine types. As we know, the recipient of BCG will develop a scar at the injection site and PV is administrated by the oral route, which is different from other vaccines. These peculiarities would give mothers a strong impression and not easily confuse with other vaccines. 

The high sensitivity and specificity of maternal recall found in this study could be due to the detail provided in the retrospective questions in the investigation, especially the initial age of every vaccine. Most of the vaccines had different initial age according to the schedule recommended by Chinese EPI. All the surveyed mothers were told specifically that HBV was given at birth, or MCV was given at eight months of age, etc. Since the data collection techniques were known to affect the level of bias [26], we suggested that more details of the vaccination be provided to the respondent where the vaccination record was not available to reduce the recall bias. There was a need to explicitly specify age limits for each vaccine in the questionnaire, seeking the accurate maternal recall for vaccine types. A previous study in developing countries had also reported both high sensitivity and specificity of parental recall across vaccines of at least 80% and 67%, respectively [15]. The high sensitivity and specificity of recall-based data indicated that the recall could largely identify the accurate vaccination status of being vaccinated and of being unvaccinated.

Our findings that older mothers and mothers with the higher parity were less likely to be subjected to recall bias for DTP and MCV was consistent with a report from Costa Rica [19]. The possible explanation was that the sample of older mothers with the experience in child bearing should be familiar with the childhood vaccines, which would reduce the risk of recall bias. It also suggested that the maternal recall on childhood vaccination should seek from older mothers or mothers with more children to improve the accuracy and reliability in future practice. The place of delivery was also found associated with the recall bias of HBV, and it was very similar to our previous findings on the determinants of the timeliness of the birth dose of HBV and other settings [28]. To our knowledge, the birth dose of HBV is required for every registered maternity hospital in China and it should be administrated within 24 h after birth. As the hospital delivery rate had reached almost 100% in Zhejiang Province in recent years, we assumed that mothers who gave birth at hospital would receive not only the birth dose of HBV, but also the basic information on childhood vaccination. The results of great accuracy on recalling DTP and MCV status among mothers with the higher education level was also line with findings from other countries [15,16,17,27]. A previous study indicated that factors such as mother’s age, education background, and economic status of the respondents, would affect the accuracy of recall [26]. Here we gave a another explanation that mothers with lower education level might have an obstacle of communicating with vaccination providers and a poor understanding of the vaccination knowledge, which could lead to a bad memory of vaccine type their children received. Migrant mothers were found as a positive factor of recall bias for DTP and MCV. Generally, migrant status had been considered as a risk factor of low coverage, poor completeness and timeliness of childhood vaccination in several previous surveys in Zhejiang Province and other areas of China [28,29,30]. Commonly, migrant people have poor social adaptation ability, which is reflected in challenges such as adapting to a new socio-cultural environment or living in poor economic conditions. These challenges would reduce the accessibility to or awareness of the utilization of the vaccination service, which influenced the accuracy of maternal recall. 

This study had important implications for vaccination program in settings where the quality of administrative report on vaccination coverage was not optimal or reliable, and the coverage survey were therefore being used. Since the maternal recall of vaccinations was basically accurate, our findings supported the supplemental use of recall data alongside record-based estimates in situations where missing records existed, to make sure the estimates more representative. Furthermore, our study also revealed that recall bias on childhood vaccinations was generally random across the surveyed mothers, with few systematic associations between bias and maternal age, parity, education background and immigration status. Maternal recall could serve as an additional resource of the coverage estimates, but great effort were still needed to maintain a high card-retention rate with complete and accurate information documented and to strengthen the administrative coverage reporting system to provide a complete time series with fewer resource compared with the coverage survey. 

There were several limitations. First, the validity of maternal recall was assessed by assuming record-based data were the reference, but these data sometimes were incomplete or inaccurate, especially for migrant children [1,4]. Second, this study focused on mothers with children aged 12–23 months and the vaccinations scheduled before 12 months of age. Since the recall bias might be large when using old children or vaccination scheduled at a later age, further validation among older children should be conducted to evaluate the recall bias affected by time interval. Third, it was unable to compare the appropriate number of dose for vaccines needed multiple doses between the two data resources, as the recall data were difficult to distinguish the dose number. Hence, the recall of BCG and MCV were much easier and more accurate to obtain compared to OPV, HBV and DPT which included three doses. Fourth, this study was conducted only in six districts, therefore the results appropriately represented the target mothers in the catchment area while might not be generalizable to the entire mothers in Zhejiang province.

## 5. Conclusions

This study evaluated the validity of maternal recall- against record-based data on childhood vaccination in Zhejiang Province, and indicated most of the vaccination status across vaccine types was accurately identified through maternal recall. The limited recall bias were associated with parity, place of delivery, maternal age, education background, and immigration status. This study supported the use of maternal recall to estimate the vaccination coverage as an alternative in the absence of record-based data.

## Figures and Tables

**Table 1 ijerph-16-00957-t001:** Two-by-two table for calculating the sensitivity and specificity.

Agreement		Vaccination Status (Record-Based)	
		Yes	No
Vaccination status (recall-based)	Yes	True positives (a)	False positives (b)
	No	False negatives (c)	True negatives (d)

**Table 2 ijerph-16-00957-t002:** Summary distribution of the characteristics of children aged 12–23 months (*N* = 648).

Variables	Level	No. (%)
Child’s gender	Male	325 (50.2)
	Female	323 (49.8)
Parity	1	413 (63.7)
	2	204 (31.5)
	≥3	31 (4.8)
Place of delivery	Hospital	597 (92.1)
	Home	51 (7.9)
Age of mother (years)	<30	443 (68.4)
	≥30	205 (31.6)
Maternal education level	<senior middle school	120 (18.5)
	≥senior middle school	528 (81.5)
Maternal employment status	Home fulltime	141 (21.8)
	Employed	507 (78.2)
Residence	Urban	329 (50.8)
	Rural	319 (49.2)
Immigration status	Resident	397 (61.3)
	Migrant	251 (38.7)
Monthly household income per capita	<3000 CNY	132 (20.4)
	3000–6000 CNY	333 (51.4)
	>6000 CNY	183 (28.2)

**Table 3 ijerph-16-00957-t003:** Evaluations of agreement between record-based and recall-based data sources (*N* = 648).

Items	BCG	HBV	PV	DTP	MCV
Agreement (%)	97.8	94.8	97.1	90.3	92.6
Maternal recall bias (%)	2.2	5.2	2.9	9.7	7.4
Over-reporting	1.7	3.1	2.2	6.6	4.7
Under-reporting	0.5	2.1	0.7	3.1	2.7
Sensitivity	95.7	95.8	95.6	91.2	93.8
Specificity	92.6	88.5	91.5	82.0	84.1
Kappa_-unadjusted_	0.90	0.83	0.89	0.72	0.79
Kappa_-adjusted_	0.93	0.92	0.95	0.83	0.88

**Table 4 ijerph-16-00957-t004:** Logistic regression model results on the risk factors of maternal recall bias across vaccine types.

Risk Factors	Level	Vaccine Type [OR(95% CI)]				
		BCG	HBV	PV	DTP	MCV
Child’s gender	Male	Ref	Ref	Ref	Ref	Ref
	Female	0.9 (0.8–1.0)	1.0 (0.9–1.0)	1.0 (0.9–1.0)	1.1 (0.9–1.1)	1.0 (0.9–1.1)
Parity	1	Ref	Ref	Ref	Ref	Ref
	2	1.1 (0.9–1.1)	1.0 (0.9–1.0)	1.0 (0.9–1.1)	1.1 (0.9–1.2)	1.0 (0.9–1.1)
	≥3	1.0 (0.9–1.0)	1.0 (0.9–1.1)	1.0 (0.9–1.1)	1.4 (1.1–1.9) *	1.5 (1.2–2.2) *
Place of delivery	Hospital	Ref	Ref	Ref	Ref	Ref
	Home	1.1 (0.8–1.1)	2.2 (1.4–2.6) *	1.0 (0.9–1.2)	1.1 (0.9–1.2)	1.1 (0.9–1.2)
Age of mother (years)	<30	Ref	Ref	Ref	Ref	Ref
	≥30	0.9 (0.7–1.0)	0.8 (0.6–0.9) *	1.0 (0.9–1.1)	0.7 (0.6–0.9) *	0.8 (0.6–0.9) *
Maternal education level	<senior middle school	Ref	Ref	Ref	Ref	Ref
	≥senior middle school	0.9 (0.8–1.1)	0.9 (0.8–1.1)	1.0 (0.9–1.1)	0.7 (0.6–0.8) **	0.8 (0.7–0.9) *
Maternal employment status	Home fulltime	Ref	Ref	Ref	Ref	Ref
	Employed	0.9 (0.8–1.0)	1.0 (0.9 –1.1)	1.1 (0.9–1.2)	1.1 (0.9–1.2)	1.0 (1.0–1.1)
Residence	Urban	Ref	Ref	Ref	Ref	Ref
	Rural	0.9 (0.8–1.1)	0.9 (0.8–1.1)	0.9 (0.9–1.0)	0.9 (0.8–1.1)	1.0 (0.9–1.1)
Immigration status	Resident	Ref	Ref	Ref	Ref	Ref
	Migrant	0.9 (0.9–1.1)	1.0 (0.9–1.1)	1.0 (0.9–1.4)	2.6 (2.0–3.5) *	2.5 (1.9–3.3) *
Monthly household income per capita	<3000 CNY	Ref	Ref	Ref	Ref	Ref
	3000–6000 CNY	1.0 (0.9–1.0)	1.0 (0.9–1.1)	1.0 (1.0–1.1)	1.0 (0.9–1.1)	1.0 (0.9–1.0)
	>6000 CNY	1.0 (0.9–1.1)	0.9 (0.9–1.0)	0.9 (0.8–1.1)	0.9 (0.8–1.0)	0.9 (0.8–1.0)

Notes: OR presented in bold were significant with * *p* < 0.05, ** *p* < 0.01, respectively. Ref: reference.

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
