# Peer review of "Validity of Maternal Recall to Assess Vaccination Coverage: Evidence from Six Districts in Zhejiang Province, China"

_ijerph, 2019, doi:10.3390/ijerph16060957_

Round 1

Reviewer 1 Report

This is an interesting study that investigates the validity of recall bias among a province in China. The authors had conducted a survey among 648 mothers and test the agreement between their memory and vaccine registries. The result suggested recall bias ranged from 2.2% to 9.7% by vaccine types. The study provided an important implications for evidence-based vaccination program.

The whole manuscript is well prepared, written in a clear structure. The statistics and tables were  well prepared.

There are few comments:

1. The coverage of the immunization system (Zhejiang provincial immunization information system) should be reported. Are eligible children well covered and recorded in the information system? As Zhejiang is a famous province in China, the internal migration population took an unignorable portion (above 20%, according to the wikipedia) that should not be overlooked.

2. Does the result apply to disadvantaged or minority population? Again, Are the recall rate remained similar for the internal migration (probably unregistered population) ?

Author Response

1. The coverage of the immunization system (Zhejiang provincial immunization information system) should be reported. Are eligible children well covered and recorded in the information system? As Zhejiang is a famous province in China, the internal migration population took an unignorable portion (above 20%, according to the wikipedia) that should not be overlooked.

Response: we added the details of Zhejiang provincial immunization information system in the introduction section and the information of migrant population was added in the methods section.

2. Does the result apply to disadvantaged or minority population? Again, Are the recall rate remained similar for the internal migration (probably unregistered population) ?

Response: the disadvantaged or minority population were not excluded intentionally in this study and the information on ethnic did not collect in this study. So, it would be difficult to apply the results to those sub-population. According to our logistic regression analyses, the migrant children was a positive risk factor for the recall bias of DTP and MCV and this result had been reported in the manuscript.

Reviewer 2 Report

I have read with great interest the manuscript titled “Validity of maternal recall to assess the vaccination coverage: evidence from six districts in Zhejiang province, China”.

In my opinion the manuscript does not requires revisions. The manuscript is understandable, the statistics used are basic, and the sampling method is well done and explained. The results are applicable in highly populated countries such as China. The limitations section contains the main issues about the validation carried out. It is a short manuscript but correctly carried out.

Author Response

no revision.

Reviewer 3 Report

General Comments

This is a well conceived and executed study in my view, providing interesting and useful insights into the dimensions of maternal recall of vaccination events. The description of limitations was comprehensive. The reference list was relevant and adequately current. The findings, although specific to a small population, none-the-less suggest that similar questions be posed more globally to determine if national household survey data on vaccination do more universally correspond with administrative and record-based data, and, as such, provide a very useful triangulation point in coverage monitoring.

Specific Comments

There are numerous minor English language edits needed with the text to improve its readability. Listed below are some examples. 

Line 38 - Replace 'rooms' with 'room'

Line 66 - Replace 'Similar' with 'A similar'

Line 91 - Suggest consider replacing 'Besides' with 'In addition'

Line 120 - Replace 'were' with 'was'

Line 129 - Add 'status' following 'socioeconomic'

Line 222 - Add 'the' following 'influenced by'

Line 228 - Replace 'known' with 'know'

Line 246 - Suggest clarify the phrase 'would be reduce to minimize'

Line 249 -Replace 'deliver' with 'delivery'

Author Response

There are numerous minor English language edits needed with the text to improve its readability. Listed below are some examples.

Line 38 - Replace 'rooms' with 'room'

Line 66 - Replace 'Similar' with 'A similar'

Line 91 - Suggest consider replacing 'Besides' with 'In addition'

Line 120 - Replace 'were' with 'was'

Line 129 - Add 'status' following 'socioeconomic'

Line 222 - Add 'the' following 'influenced by'

Line 228 - Replace 'known' with 'know'

Line 246 - Suggest clarify the phrase 'would be reduce to minimize'

Line 249 -Replace 'deliver' with 'delivery'

Response: thank you for your comments and all the grammar errors have been fixed.